# Antiepileptic Effect of Neuroaid^®^ on Strychnine-Induced Convulsions in Mice

**DOI:** 10.3390/ph15121468

**Published:** 2022-11-26

**Authors:** Ahmed Salim Mahmood, Afaq M. Ammoo, Mayssam Hussein Mohammed Ali, Tiba M. Hameed, Hany A. Al-Hussaniy, Abdulla Amer Abbas Aljumaili, Mohammed Hussein Alaa Al-Fallooji, Ali Hakim Kadhim

**Affiliations:** 1Department of Pharmacy, Al-Rasheed University, Baghdad 10001, Iraq; 2Department of Pharmacy, Bilad Alrafidain University College, Diyala 32001, Iraq; 3Baghdad Teaching Hospital, Medical City Directory, Baghdad 10001, Iraq; 4College of Medicine, University of Baghdad, Baghdad 10001, Iraq

**Keywords:** NeuroAid, epilepsy, glutamate receptor, gene expression, topiramate, strychnine

## Abstract

NeuroAid II, a folk Chinese Medicine, is currently used in Asia for the treatment of stroke. An experimental study demonstrated that NeuroAid enables neuronal cells to be more resistant to glutamate toxicity. This research was constructed to evaluate the efficacy of NeuroAid in the prevention of epilepsy (EP). Forty healthy adult male mice were used and divided into four groups (10 mice/group): normal control group; positive control group; NeuroAid-treated group (10 mg/kg); topiramate-treated group (10 mg/kg). The treatment continued for 7 days, and on the last day, EP was induced using strychnine at a dose of 2 mg/kg via intraperitoneal (ip) administration. Seizure severity, latency to the seizure onset, the number of seizures, and the duration of each seizure episode were observed for one hour. The death and protection rates over the next twenty-four hours were recorded. Brain specimens from surviving animals were extracted and examined pathologically for quantification of glutamate receptor (GluR) gene expression in the isolated hippocampus employing real-time PCR analysis. Treatment with NeuroAid resulted in a significant reduction in seizure severity, prolonged the onset of seizures, decreased the number and duration of episodes, reduced brain insult, and decreased mortality rate. Reductions in the gene expression of GluRs in the hippocampus with minor histopathological changes were observed in the NeruoAid- and topiramate-treated groups. It is concluded that NeuroAid has a potential antiepileptic effect (EP) with the ability to prevent convulsion through its effect on the glutamate receptor.

## 1. Introduction

Epilepsy (EP) is a collection of pathological disorders in the brain characterized by epileptic seizures. Epileptic seizures are episodes that can vary from brief and nearly undetectable periods to long periods of vigorous convulsions. These episodes can result in physical injuries, occasionally including broken bones. In epilepsy, the seizures tend to recur and, as a rule, have no immediate underlying cause. People with EP may be treated differently in various areas of the world and experience varying degrees of social stigma due to their condition [1,2,3]. Unfortunately, current drugs have a wide range of limitations. They provide relief in only up to 75% of patients with absence seizures, in up to 85% of patients with generalized tonic-clonic seizures, and in 65% of patients with new-onset seizures. Seizure recurrence occurs in 5% of those with EP, and 35% have uncontrolled EP. Tolerance and loss of efficacy occur with drug use over a period of time, with a possible risk of drug interactions, and approximately 30% of patients with EP show drug resistance, which is the major problem associated with antiepileptic drugs [4,5]. Accordingly, studies have continued to identify new drugs that effectively prevent EP with minimal side effects and fewer drug-drug interactions. Most of these drugs target glutamate receptors.

There are two types of glutamate receptors that have different physiological functions. The first one is the ionotropic receptor, which includes the alpha-amino-3-hydroxy-5-methyl-4-isoxazole-propionic acid (AMPA), *N*-methyl-d-aspartate (NMDA), and kainite receptors. The second type is the metabotropic receptors that act through second messenger systems. Four subtypes for AMPA receptors designated as GluR1, GluR2, GluR3, and GluR4 have been identified. The GluR1 subunit increases the conductance of calcium ions during synaptic transmission [6]. It is a recognized fact that excessive neuronal stimulation mediated by GluR is one of the pathological factors involved in the development of seizures. The most widely targeted GluRs that play a vital role in the pathogenesis and neuronal physiology of brain diseases are the NMDA and AMPA receptors [7]. Seizures can be induced in animals by NMDA agonists; conversely, experimental animal studies have demonstrated the ability of NMDA antagonists to prevent seizure induction, suggesting a potential role for NMDA receptor antagonists in antiepileptic drug research [8]. Many drugs have been evaluated in clinical trials; however, the majority of NMDA receptor antagonists have failed to exert good efficacy and a good safety profile for the treatment of epilepsy. NeuroAid (MLC 901; Moleac Pte Ltd., Singapore) is a Chinese herbal medicine that is widely used in the treatment of stroke in Asia [9]. In vitro and in vivo results show that NeuroAid makes cells more resistant to glutamate toxicity, increases neurite outgrowth and connectivity, and reduces infarct volume [10]. For these reasons, this research was conducted to evaluate the efficacy of NeruoAid in the prevention of EP using a strychnine-induced convulsion (SIC) animal model and to compare its efficacy with topiramate.

## 2. Results

### 2.1. Effect of NeuroAid on SIC

Following the induction of seizures, animals showed variations in the incidence of seizures, time of onset, duration of episodes, and frequency of convulsion. As shown in Table 1, NeuroAid at a dose of 10 mg/kg/day significantly increased the latency period to 6.92 ± 2.99 min when compared to the positive control group, which showed induction of EP after 4.28 ± 1.64 min. Similarly, mice that received topiramate (reference drug) also had a significantly delayed onset of convulsions compared with the positive control group (*p* < 0.05) but with a higher value (8.66 ± 5.24) when compared with mice that received NeuroAid. The duration of convulsions in the NeuroAid-treated group and the topiramate-treated group were reduced significantly as well, with calculated means of 8.16 ± 9.53 s and 6.68 ± 6.61 s, respectively, as compared to animals subjected to strychnine only at 25 ± 7.07 s (*p* < 0.05). Furthermore, Table 1 also displays the mean frequencies of convulsion after induction. The number of convulsions was significantly reduced (*p* < 0.05) in the NeuroAid-treated group and topiramate-treated group when compared with positive control group, they showed no significant variation between them regarding this parameter (*p* > 0.05). The study also included the calculation of the incidence of convulsion in each group 1 h after induction and the mortality rates in study groups after 24 h. Table 2 illustrates that the mortality rate of mice treated by NeuroAid was 40%, which was comparable to the topiramate-treated group (30%). These results statistically differed from the positive control group, which showed the highest percentage of death (100%) after 24 h of induction with strychnine. These results imply that treatment with NeuroAid can provide 60% protection against SIC, while the protection rate for topiramate was 70%.

### 2.2. Effect of NeuroAid Treatment on Gene Expression

In the NeuroAid-treated group and topiramate-treated group, there was a significant reduction in the expression of the GluR1 mRNA in the hippocampus when compared with the positive control group (*p <* 0.05). An increase in gene expression in the GluR1 subunit was observed in the positive control group when compared with the normal control, non-seizure group *(p <* 0.05), as shown in Figure 1.

### 2.3. Effect of NeuroAid on Histopathological Examination

A histopathological examination was performed on the rat brains in each group. The brain sections from the positive control group (Figure 2B) showed that the pyramidal cell layers comprised loosely packed, degenerated, atrophied, and vacuolated neurocytes. The cell nuclei were shrunken, pyknotic, and hyperchromatic. Furthermore, degeneration was seen in the astrocytes and microglia. On the other hand, minor changes were seen in the NeuroAid- and topiramate-treated groups (Figure 2C,D). These changes were represented by diffuse vacuolation, some neuronal damage, and a few apoptotic astrocytes, with improvements in the neuronal structure.

## 3. Discussion

Epilepsy is considered the second most prevalent brain disease after stroke and has a strong effect on the health of the entire human body system [11]. Currently available AE treatments have numerous adverse effects and exhibit pharmacoresistance. Consequently, it is essential for scientists to discover newer anticonvulsant drugs having wide safety profiles comparable to current drugs with enhanced pharmacological effects because EP requires long-term treatment [12]. Traditional medicinal plants have moderate bioreactivity; hence, they are considered safer than synthetic drugs. Numerous studies are currently examining the active constituents of medicinal plants, allowing their utilization in the development of new drugs [13]. In this research, the anticonvulsant effect of NeuroAid was evaluated in animals using a SIC model, which produces convulsions through interference with postsynaptic inhibition of spinal cord neurons facilitated by glycine (an inhibitory neurotransmitter). Additionally, the levels of the amino acid, glutamic acid in the brain are also increased by strychnine, which acts as a neurotransmitter for excitatory nerve impulses, leading to muscle contractions [14]. The main finding in this study was that the administration of NeuroAid to mice for 7 days significantly delayed the onset of seizures and decreased their frequency and duration. The efficacy of NeuroAid was comparable to topiramate. In addition, the tested drug resulted in an 80% incidence of EP with 60% protection from death within 24 h. NeuroAid consists of a mixture of nine diverse herbal extracts in one capsule (radix astragali, radix salvia mitorrhizae, radix paeoniae rubrae, rhizome chuan-xiong, radix angelicae sinensis, Carthamus tinctorius, Prunus persica, radix polygalae, and rhizome acori tatarinowii) [15]. The main components of these plant extracts are alkaloids, phthalides, organic acids, polysaccharides, flavones, coumarin, glycosides, flavonoids, saponins, and carbohydrates [16,17,18,19]. The antiepileptic effect of some flavonoids and glycosides has been evaluated. The isolation and identification of more than 5000 flavonoids have been achieved, and their mechanisms of action also have been identified. The mechanism of action underlying the antiepileptic effects of the identified flavonoids is through GABA chloride channels, which may be due to the structural similarity between benzodiazepines and these flavonoids [20,21]. This may explain the role of NeuroAid in the prevention of epilepsy. Their antioxidant properties may be the second reason for the anticonvulsant effect of NeuroAid. Research has revealed that seizures attenuate the antioxidant protection afforded by the brain as well as the generation of free radicals; consequently, this further provokes oxidative stress, lipid peroxidation, brain edema, and epilepsy. Furthermore, the production of the excitatory neurotransmitter glutamic acid is increased due to the inactivation of glutamine synthase by the high levels of free radicals, resulting in seizure induction. Another enzyme inhibited by free radicals is glutamate decarboxylase, which leads to a decrease in GABA levels in the brain [22]. Molecular docking was also used in another study that evaluated the effectiveness of resveratrol (a strong antioxidant) in the prevention and progression of myoclonus epilepsy [23]. Quintard and colleagues found that the use of NeuroAid II was associated with significant reductions in Bax expression and the level of malondialdehyde (a product of lipid peroxidation) [24]. Extracts from Acorus tatarinowii (one of the components of NeuroAid) were also shown to exert antiepileptic effects in mice. The compound contains two types of essential oils, including asarone, which has been shown to afford neuroprotection by mitigating oxidative stress and neuroinflammation, as well as promoting neuronal cell survival [25]. The administration of NeuroAid in drinking water (6 mg/mL) for one week showed positive changes in the histopathological structure of the hippocampus [10]. Heurteaux et al. (2010) correlated this improvement to the antiapoptotic role of NeuroAid and its ability to maintain the structural integrity of mitochondria with protective effects against excitotoxic damage [10].

Our research suggests another mechanism for the effect of NeuroAid in the treatment and prevention of epilepsy. The data collected from the real-time-PCR analysis revealed that NeuroAid has the ability to decrease the expression of the GluR1 subunit in the hippocampus. Glutamate is considered a highly abundant excitatory neurotransmitter in the brain and spinal cord, which upon release into the synaptic space, will bind to GluR, resulting in the generation of an action potential under physiological conditions. Many factors play a role in the incidence of epilepsy. Excessive glutamatergic neurotransmission is considered one of the underlying factors of EP [26]. These findings are augmented by previous in vitro and in vivo results, which have demonstrated the ability of NeuroAid to potentiate neuronal cells and prevent glutamate toxicity [9].

There are two major limitations in this study that could be addressed in future research. First, the study focused on a strychnine-induced convulsion model, and there are many models for the induction of epilepsy, such as those using pentylenetetrazole, pilocarpine, and electroshock-induced seizures. These models can be used to exclude other supposed mechanisms for the tested drug. Second, as mentioned above, some studies showed that there is a correlation between oxidative stress and the incidence of epilepsy. Furthermore, NeuroAid contains in its constituents some compounds that have antioxidant properties, and as such, there is a need for measuring oxidative biomarkers in brain tissue, such as superoxide dismutase and catalase antioxidant enzyme activities, as well as malondialdehyde levels, as a sign of lipid peroxidation.

## 4. Materials and Study Design

### 4.1. Drugs and Chemicals

NeuroAid (MLC 901, Moleac Pte Ltd., Singapore), topiramate (TOPAMAX^®^ Janssen, Antwerpseweg 15, Beerse, Belgium) sodium chloride 0.9% (Pioneer, Sulaymaniyah, Iraq) and Strychnine (Sigma, St. Louis, MI, USA) were all used in the experiment.

### 4.2. Experimental Animals

Forty healthy BALB/c male mice weighing approximately 25–30 g were obtained from the animal breeding house at the Center of Quality Control and Drug Research, Baghdad, and transported for the experimental study to the animal house in the Department of Pharmacy, Al-Rasheed University College, Baghdad. The animals were kept in a temperature-controlled environment (22–25 °C) under a 12 h light/12 h dark cycle. The humidity range of the air was 25–30%, allowing the animals free access to drinking water and standard mice food. The animal care protocols were approved by the Ethics Committee of the Al-Rasheed University College, Department of Pharmacy (Number 11 on 10 November 2021). One week after acclimatization, the mice were divided into four groups (10 mice for each group).

### 4.3. Preparation of NeuroAid Drinking Solution

In this experiment, we prefer to give the drugs in drinking water for two reasons. First, this was to avoid stressful situations induced by oral gavage, which may lead to increases in blood pressure, heart rate, and cortisol, which may have an effect on the final results [27]. Second, this was to avoid missing a dose of NeuroAid because the drug is usually given three to four times daily [28]. The drinking solution was prepared as prescribed Heurteaux et al. (2010) by dissolving one capsule of MLC901 in 66 mL of water using a magnetic stirrer at 37 °C for one hour. Then, the solution was added to the drinking water at a concentration of 6 mg/mL [10].

### 4.4. Preparation of Topiramate Solution

Topiramate was given to the mice at a dose of 10 mg/kg [29]. The drug was delivered to the animals in drinking water as well. According to the study performed by Bachmanov et al. (2002), the daily water intake for BALBc strain mice is 5.7 mL ± 0.2 mL. We assumed that the average daily intake for each mouse was 6 mL/30 g body weight, so we prepared the topiramate solution by crushing 2 tablets of 25 mg topiramate and dissolving them in 1000 mL of drinking water (the solution was freshly prepared every day) [30].

### 4.5. Study Design

The mice in this study were categorized into four groups (10 mice per group): normal control group—the mice were left without treatment and considered as a negative control group injected with normal saline intraperitoneally; positive control group—the mice received strychnine only (2 mg/kg, intraperitoneally) for the induction of EP and were left without treatment; NeuroAid-treated group—the mice were pretreated with NeuroAid at a dose of 10 mg/kg received orally in drinking water; topiramate group—the mice in this group were pretreated with topiramate (reference standard drug) at a dose of 10 mg/kg delivered orally in drinking water at 10 mg/mL.

### 4.6. Induction of Epilepsy

Epilepsy was induced by strychnine, as described by Annafi et al. in 2014 [31]. Strychnine was dissolved in normal saline one day before induction, then passed through a sterile mill polar filter to be ready for use the next day. After 7 days of ‘drug’ treatment, strychnine was injected intraperitoneally at a dose of 2 mg/kg of body weight for all mice in each group, with the exception of the negative control group. Following induction, the behavior of each mouse was recorded for 1 h using a recording camera and observed for seizure changes, the time of onset of clonic seizures, and the duration of, and the number of convulsion episodes was recorded by placing the mouse in a separate cage. Many models of EP induction are available, but the reason we chose this model was due to the ability of strychnine (a glycine receptor antagonist) to suppress postsynaptic Ca^2+^ influx, suggesting greater potentiation of the NMDA receptor by the glycine receptor compared to GABA(A) receptors [32,33].

### 4.7. Haematoxylin and Eosin (HE) Staining

After 24 h of EP induction, the animals were sacrificed, and the brains were dissected on an ice plate to remove the hippocampus by decapitation. The brains of 3 mice from each group were removed rapidly, washed with normal cold saline, dissected to remove the hippocampus from one half of the brain, kept in TRIzol extraction solution, and frozen at −20 °C until the real-time PCR analysis was done. The second half of each brain was kept in 10% formalin for histopathological examination. This part of the brain was then embedded under optimal cutting temperatures. After this, a tissue slicer was used to obtain slices measuring six micrometers thick on a microscopic slide, which was then stained using HE solutions for morphological examination under the light microscope.

### 4.8. Extraction of RNA

The extraction of RNA is the first step in the detection of genes in tissue samples. The procedure starts with the lysis of the tissue in order to free the nucleic acid from the cells. This is conducted by adding cell pellets to the lysis buffer, which contains reagents such as quinidine and Triton X that destroy cells, denature protein, inactivate RNase and create appropriate conditions that favor the adsorption of RNA to the silica membrane. Then, additional disruption is achieved using a MagNA lyser (Roche, Rotkreuz, Switzerland) in 2 intervals of 6500 r.p.m. for 60 s with an intermediate cooling step for one minute on ice. DNA digestion is done, and only total RNA remains in the filter to be isolated using the RNeasy mini kit (Oiagen) according to the manufacturer’s instructions.

### 4.9. Real-Time PCR Analysis

A real-time PCR analysis was performed to measure the gene expression of GluR in the hippocampus. The analysis was performed using a 480-light cycler (Roche company). The reactions were performed using a total volume of 10.1 L, containing 4.8 ng cDNA template (with an exception for 16S rRNA gene primers, for which 0.48 ng of cDNA was used). In addition to this, an optimized primer concentration (Table 3) and light cycler, and a 480 SYBR Green I master mixer was used. The controls were water without a template and RT without samples. Specific PCR cyclic conditions were used, which included 8 min of heat starting at 95 °C and 45 amplification cycles (95 °C for 10 s, 57 °C for 15 s, 72 °C for 20 s, 78 °C for 1 s with a single fluorescence measurement). The melting curve (60–95 °C at 2.2 °C for 1 s) was continued for the measurement of fluorescence. The final step was cooling. Peaks were observed in the melting curve analysis, and the single peaks confirmed the specificity of the amplification. On the other hand, standard curves, which depended on the genomic DNA, were generated to determine the efficiency of the housekeeping gene’s target amplification via RT-PCR. Two separate sets of experiments for all reactions were performed, which involved triplicates for each sample, and the mean values were used for the results. The expression levels of the housekeeping gene were compared using crossing points.

### 4.10. Statistical Calculation and Analysis

The results obtained were analyzed statistically via a one-way ANOVA using SPSS version 24 and Graph Pad Prism Software version 9 for the figure presentation. The data are expressed as means ± standard deviation (SD). The level of *p* < 0.05 was considered statistically significant.

## 5. Conclusions

The present experimental study explored the potential role of NeuroAid as an anticonvulsant agent by reducing the frequency and duration of convulsion attacks and the mortality rate. The suggested mechanism for the antiepileptic effect is a reduction in the gene expression of GluR in the hippocampus. For this reason, we suggest the isolation of the active constituents responsible for the neurological protection exerted by NeuroAid or its use as a supplement with other antiepileptic drugs at decreased doses to minimize their side effects.

## Figures and Tables

**Figure 1 pharmaceuticals-15-01468-f001:**
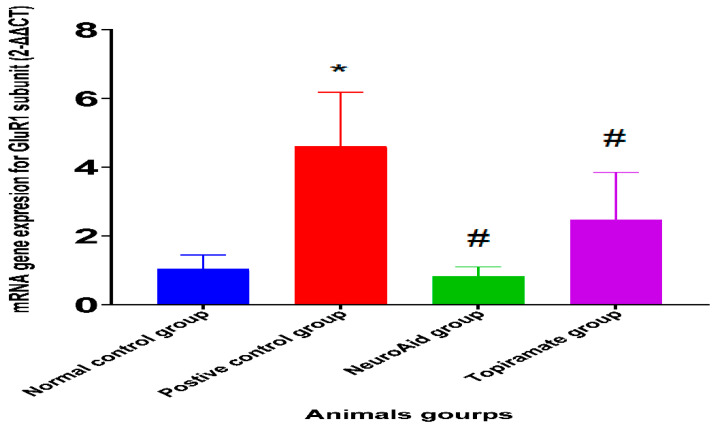
Effect of NeuroAid treatment on the level of GluR1 subunit gene expression (mean ± SE) using real-time PCR analysis compared to the topiramate-treated group. * (*p <* 0.05)—significant difference between positive control and negative (normal) control group. # (*p <* 0.05)—significant difference between treated groups compared to the positive control group.

**Figure 2 pharmaceuticals-15-01468-f002:**
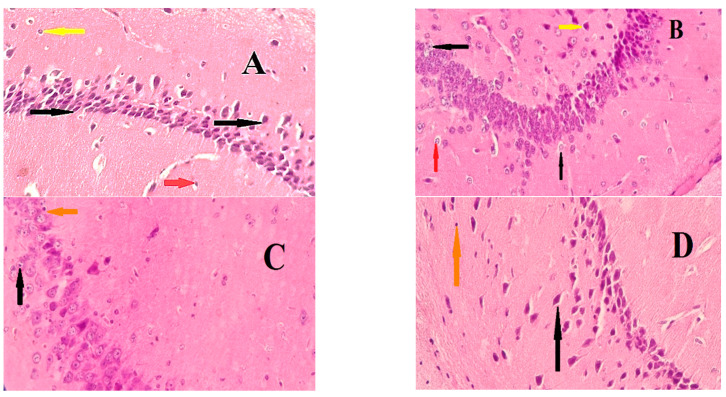
Microscopic sections of the rat hippocampus: (**A**) section from the normal control group, showing a regular cellular architecture with pyramidal cells (black arrow), astrocytes (red arrow) and microglial cells (yellow arrow); (**B**) section from the positive control group, demonstrating degenerated neurons with shrunken nuclei (black arrow), numerous hyperchromatic astrocytes with a vacuolated cytoplasm (red arrow) and hyperchromatic microglial cells (yellow); (**C**,**D**) normal pyramidal cells with some neurons that are pyknotic (black arrows) and astrocytes with a vacuolated cytoplasm (orange arrow).

**Table 1 pharmaceuticals-15-01468-t001:** Effects of NeuroAid on convulsion parameters induced by strychnine.

Animals Group	Onset of Convulsion (min)	No. of Convulsions (per One h)	Duration of Convulsion (s)
Normal control group	00 ± 00	00 ± 00	00 ± 00
Positive control group	4.28 ± 1.64	2.67 ± 1.03	25.00 ± 7.07
NeuroAid group	6.92 ± 2.99	1 ± 1.09 *	8.16 ± 9.53 *
Topiramate group	8.66 ± 5.24	1 ± 0.63 *	6.68 ± 6.61 *

Note: *n* = 10, values are stated as means ± standard deviation of the mean; * *p* ≤ 0.05 values are significant in contrast to the positive control group.

**Table 2 pharmaceuticals-15-01468-t002:** Effects of NeuroAid and topiramate on the mortality rate and protection of mice.

Animals Group	No. of Animals		Mortality Rate (%)	Protection Rate (%)
	Dead	Alive		
Normal control group	0	10	0%	100%
Positive control group	10	0	100%	0%
Neuro Aid group	4	6	40% *	60% *
Topiramate group	3	7	30% *	70% *

Note: *n* = 10 values are stated as means ± standard deviation of the mean; * *p* ≤ 0.05 values are significant in contrast to the positive control group using chi-square analysis.

**Table 3 pharmaceuticals-15-01468-t003:** Specific gene primers are utilized in the real-time PCR.

Primers Type	5′-3′ Gene Sequence	Length (bp)	Amplificated Fragment (bp)
GluR1 FP	CAGATCGATATTGTGAACATCA	22	400
GluR1 RP	CCTGAAAGAGCATCTGGTAT	20	
β-Actin FP	TACCAACCTCCTTGCAGCTCC	20	800
β-Actin RP	ACAATGCCGTGTTCAATGG	19	

FP, forward primer; RP, reverse primer.

## Data Availability

Data is contained within the article.

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
