# Peer review of "Antiepileptic Effect of Neuroaid® on Strychnine-Induced Convulsions in Mice"

_pharmaceuticals, 2022, doi:10.3390/ph15121468_

Round 1

Reviewer 1 Report

The manuscript provides interesting information, nevertheless some aspects should be addressed, i.e. please provide information about mice sex (males/females). An interesting addition to the study could be to investigate the oxidative status of brain tissues and the level of lipid peroxidation?

Reviewer 2 Report

In the present study, the authors demonstrated an anticonvulsive effect of NeuroAid in a mouse model of seizure induced by strychnine. Several measures including seizure onset, duration, number of seizures, and mortality were assessed. Additionally, it was found that the seizure-induced mRNA overexpression of hippocampal GluR1 as well as brain histopathological changes were reduced by NeuroAid and also topiramate as control anti-seizure medication. Overall, the study is well designed and well written. I have few minor recommendations to improve the quality of paper.

Comment 1. An explanation about different glutamate receptors (especially, NMDAR and AMPA) and their subtypes in the CNS and their role in epilepsy is necessary in the Introduction. This helps readers understand better the rationale behind choosing GluR1 as target receptor in this study as well as they would know which glutamatergic receptors GluR1 belongs to!

Comment 2. The abbreviation for glutamate receptor should be either GluR or GlutR throughout the manuscript, not both. Please unify this in all manuscript.

Comment 3. If it is not against the journal guideline for authors, it would be better to have material and methods section prior to results section.
